# Adaptive Noise-Resistant Low-Power ASK Demodulator Design in UHF RFID Chips

**Yao-Hua Xu** 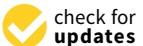**, Shuai Yang** , **Hang Li, Ji-Ming Lv and Na Bai** *

School of Integrated Circuits, Anhui University, Hefei 230601, China; xyh@ahu.edu.cn (Y.-H.X.);
p21201049@stu.ahu.edu.cn (S.Y.); p21301249@stu.ahu.edu.cn (H.L.); p21301248@stu.ahu.edu.cn (J.-M.L.)
* Correspondence: 11074@ahu.edu.cn

**Abstract:** This paper presents a new signal demodulator for ultra-high frequency (UHF) radio frequency identification (RFID) tag chips. The demodulator is used to demodulate amplitude shift keying (ASK) modulated signals with the advantages of high noise immunity, large input range and low power consumption. The demodulator consists of a charge pump, an envelope detector, and a comparator. In particular, the demodulator provides a hysteresis input signal to the comparator through two envelope detectors, resulting in better noise immunity. The demodulator is based on a standard 0.13 μm CMOS process. The demodulator is suitable for demodulating high frequency signals at 900 MHz with a data rate of 128 Kbps and can operate up to 78 °C. The input signal has a peak of 1.2 V and consumes as little as 113.6 nW. The demodulator also has a noise immunity threshold of approximately 3.729 V.

**Keywords:** ASK; demodulator; RFID; hysteresis of input; differential operational amplifier



## 1. Introduction

The demodulator is used to reduce the high-frequency modulated signal to a low frequency baseband signal. Demodulators are often used in applications such as RFID chips, electronic tags, telephone lines, etc., mostly for high-frequency signal reduction or digital-to-analog conversion [1–3].

In [4], the authors forgo the use of large capacitors in their design in exchange for a small chip area and a good implantable chip experience, yet suffer from design complexity and high-power consumption. In [5], the authors claim that their circuit can operate with a minimum modulation index (MI) of 5.26% and a speed of 0.5 Mb/s by using shifters and switches. Thus, the design can be well used for biomedical implants, but is not suitable for UHF applications due to the low switching speed. In [6], the authors used a hysteresis input structure design to reduce the noise impact and a secondary amplifier to increase the input signal range, but it still has the disadvantage of high-power consumption. In [7], the authors improve the efficiency of the voltage conversion from RF port to DC interface (RF–DC) by threshold voltage compensation, leakage current suppression and dynamic voltage adjustment schemes, but consume a larger chip area.

To overcome the above-mentioned disadvantages, the adaptive noise-resistant low-power ASK demodulator design implemented in this paper has the following outstanding advantages:

1. It has the advantage of adaptive noise reduction and can still work effectively under high noise conditions.
2. The comparator section, designed with two differential op-amps, shares a common bias circuit, and the outputs are directly coupled to reduce demodulation errors caused by internal circuit heat generation.
3. Compared with the data given in the references, the power consumption value of this design is about one hundredth of the power consumption value given in the

references [4–6] and the power consumption value of this design is about one third of the power consumption value given in the reference [7]. This paper gives specific data comparisons in Section 3.2 [8–10].

This paper is organized as follows. Section 2 describes the role of demodulator in the communication system, the general structure of UHF RFID, and the design details of each module of the demodulator. Section 3 discusses the simulation results of the demodulator in terms of noise immunity and low power consumption. Section 4 concludes the paper.

## 2. Methods

### 2.1. Digital Communication System with Passive UHF RFID Tag Structure

Since the signals received by RFID tags are transmitted in a wireless channel and are affected by noise, this issue needs to be considered when designing the demodulator. The flow chart shown in Figure 1 illustrates the whole communication process and shows the role played by the demodulator in the communication system. In other words, the entire communication process relies on the demodulator to demodulate the high frequency signal in order to perform subsequent processing, thus enabling communication between the transmitter and receiver [11].

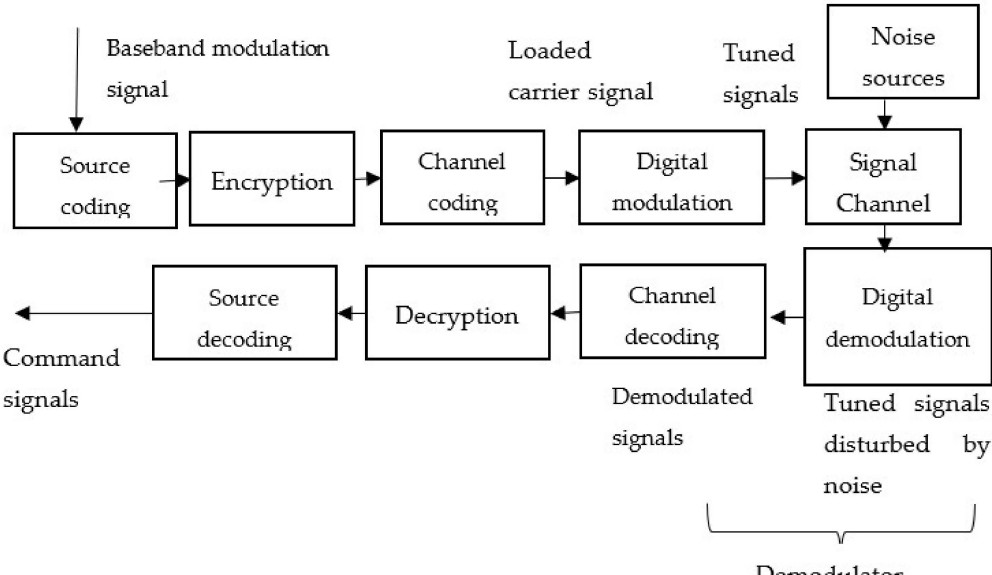

**Figure 1.** Function of the demodulator in the digital communication system.

The RFID tag consists of an analog front-end, a digital state machine, and an off-chip antenna. The analog front-end consists of a rectifier, a voltage regulator, a demodulator, a modulator, and a reset circuit, as shown in Figure 2. When the RFID reader sends a command to the RFID tag, the RFID tag acts as the receiver of the digital communication system.

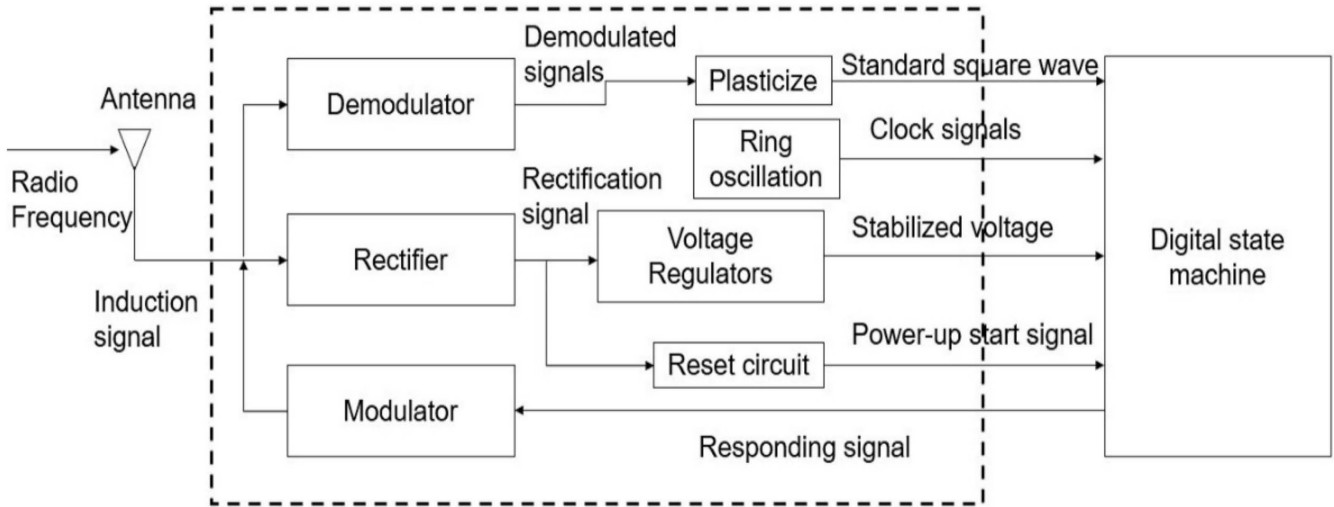

**Figure 2.** Block diagram of passive UHF RFID tags.

The 10 m long-range operating range of UHF RFID is one of the main reasons it is widely used, and the following is the theoretical basis for the long-range transmission of UHF RFID [12]. The reading distance of the reader, i.e., the working ranger of the reader and RFID tag, can be derived from Equation (1):

$$r = \frac{\lambda}{4\pi} \sqrt{\frac{EIRP \cdot G_r \cdot \tau \cdot p}{P_h}} \tag{1}$$

where $\tau$ is the mismatch factor, $EIRP$ is the equivalent omnidirectional radiated power, $p$ is the polarization loss factor, $P_h$ is the code piece threshold power, $G_r$ is the tag antenna gain, and $\lambda$ is the free-space wavelength. The mismatch factor $\tau$ and polarization loss factor $p$ can be 0.5, $EIRP$ can be obtained from Equation (2). The tag antenna gain $G_r$ can be obtained from Equation (3). $\lambda = C/f$, $C$ is the speed of light, and $f$ is the operating frequency. The code piece threshold power $P_h$ determines the lower power at which the tag works, namely, the tag can work properly only when the input power exceeds $P_h$. The formulas for $EIRP$ and $G_r$ are given below:

$$EIRP = (P_T - L_c)G_a \tag{2}$$

$$G_r = \frac{4\pi A_e}{\lambda^2} \tag{3}$$

where $P_T$ is the transmit power, $G_a$ is the antenna gain, $L_c$ is the loss on the feed line, $\lambda$ is the free space wavelength, and $A_e$ is the effective area of the antenna.

For ASK demodulators, the envelope detector is an important component to achieve its demodulation function [13,14]. The following is the rationale for the envelope detector to demodulate the baseband signal from the UHF signal (860–960 MHz). The envelope detector model is shown in Figure 3. The RC limit formula is given below:

$$\frac{1}{w_0 C} \leq R \leq \frac{1}{wC} \tag{4}$$

where $R$ is the subsequent circuit resistance, $w$ is angular velocity of the local carrier, $C$ is the capacitance capacity, and $w_0$ is the baseband signal frequency.

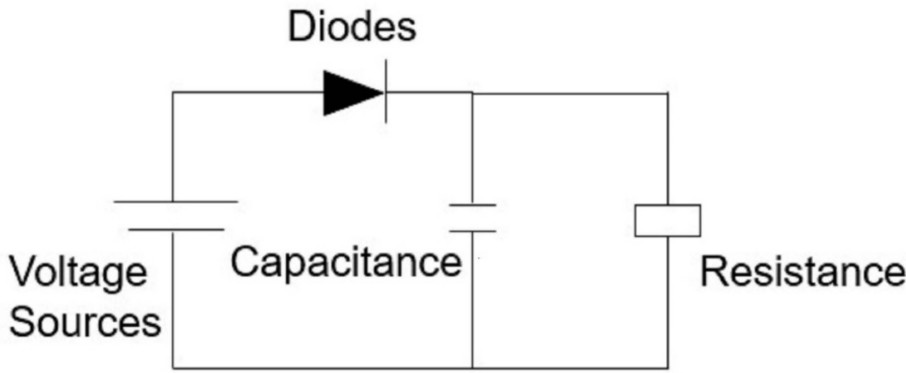

**Figure 3.** Working model of the envelope detector.

For the envelope detector, improper settings of the resistor *R* and capacitor *C* can produce inert distortion. Specifically, if the product of *C* values is too large, it will result in too large gap between charge and discharge, thus making the output signal change too slowly. To prevent this phenomenon, the limiting equation is given below:

$$\frac{5 \sim 10}{\Omega} \le RC \le \frac{\sqrt{1 - m_a^2}}{m_a \Omega} \tag{5}$$

where *R* refers to the envelope detector subsequent circuit total resistance value, and *C* refers to the capacitor resistance, $\Omega$ refers to the input signal frequency, and the $m_a$ for the amplitude modulation coefficient. $\sqrt{1 - m_a^2}/m_a$ usually takes the value 1.5.

At the end, a simple power consumption calculation formula is given below to meet the calculation requirements in this paper. The formula is shown as follows:

$$P = UI \tag{6}$$

where *P* is the average power, *U* is the effective voltage, and *I* is the average current. Since this simulation uses a sine wave with a peak value of 1.2 V as the input signal, the value of the effective voltage *U* is the carrier peak value of $1.2/\sqrt{2}$ V. The transient current waveform at the input of the charge pump is then measured, and then the rms calculation is performed by the internal calculator of cadence to obtain the average current.

The demodulator plays a crucial role in RFID tags. Modulator noise immunity directly determines the correctness of the command received by the digital state machine and whether it can work adequately. Demodulator common-mode input range, which determines the wide range of demodulator applications in the market. For passive RFID, the RFID tag functions by extracting energy from the received RF signal.

### 2.2. Low-Power ASK Demodulator Design with UHF Adaptive Noise Immunity

The UHF adaptive noise-resistant low-power ASK demodulator consists of three modules: a CMOS low-threshold charge pump, an adaptive noise-resistant envelope detector, and a high common-mode input, dual-differential op-amp hysteresis input comparator. This design adopts the adaptive anti-noise design idea. The following section gives the demodulator workflow. The tuned signal is initially demodulated by a charge pump in order to demodulate the baseband signal from the 900 MHz UHF carrier signal. The obtained baseband signal is passed through the envelope detector to obtain the hysteresis signal of the preliminary demodulated signal. The preliminary demodulated signal and the hysteresis signal are passed through the comparator, respectively to obtain the square wave corresponding to the baseband signal by differential comparison. The framework diagram of the demodulator is shown in Figure 4.

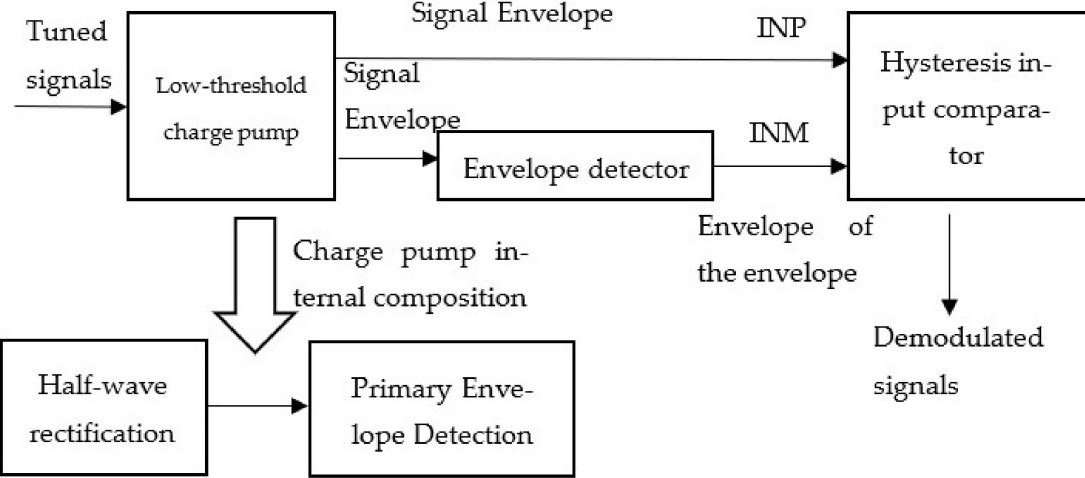

**Figure 4.** Block diagram of the ASK demodulator.

The CMOS low-threshold charge pump, adaptive noise-resistant envelope detector, high common-mode input, and dual-differential op-amp hysteresis-input comparator are described in detail in the following sections, respectively.

### 2.2.1. CMOS Low-Threshold Charge Pump

The CMOS low-threshold charge pump is used for initial demodulation of the modulated signal. The CMOS low-threshold charge pump of this design consists of two parts. The half-wave rectifier module, composed of PM0 and capacitors C0, performs half-wave rectification on the received tuned signal to filter out the reverse waveform part for subsequent demodulation. Envelope detector module, composed of PM1 and capacitors C1, performs preliminary demodulation of the tuned signal and filters out the high-frequency part, as shown in Figure 5. In Figure 6, INPUT is the charge pump input signal, obtained by loading a baseband signal with a peak of 1.2 V onto a 900 MHz carrier, and OUTPUT is the charge pump output signal with a period of 5 μs and a peak above 170 mV. As can be seen, the demodulated signal has the same period as the baseband signal, and the peak is the same as the baseband signal, i.e., the charge pump has performed the initial demodulation.

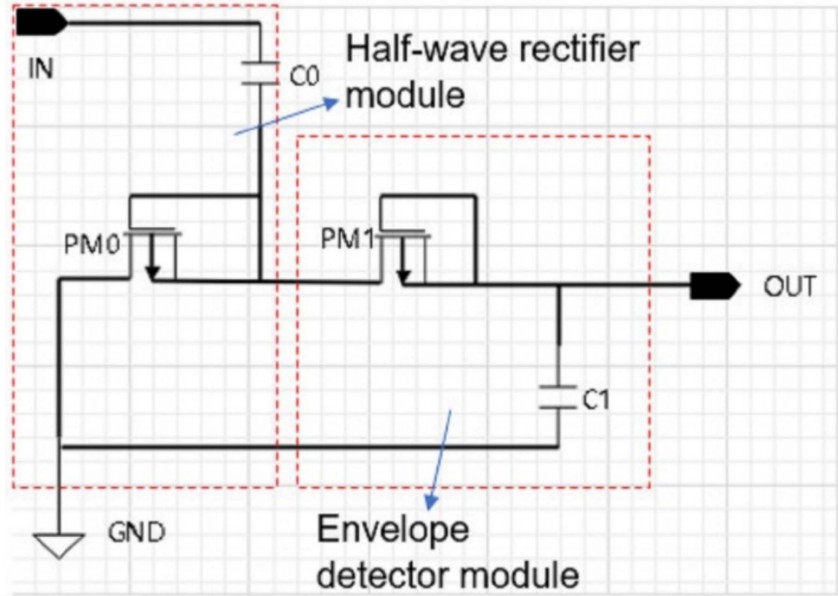

**Figure 5.** CMOS low threshold charge pump.

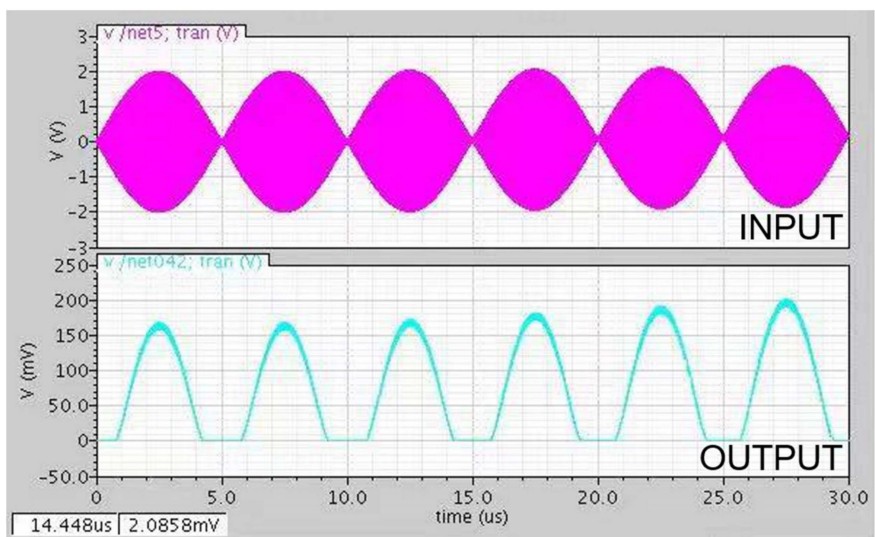

**Figure 6.** Charge pump simulation results.

The body terminals of the PMOS transistor of this design are connected to the gate and drain terminals (reverse bias) to reduce the effective threshold voltage and power consumption (in this design, when the body terminal is connected to the source, the threshold voltage is −381.742 mV. However, when the body terminal is connected to the drain and gate, the threshold voltage becomes −303.397 mV). Then, the high-frequency signal is half-wave rectified by the envelope detector composed of PMOS and capacitor. The envelope detector extracts the high-frequency modulated signal envelope and initially demodulates it to obtain the low-frequency demodulated signal.

2.2.2. Adaptive Anti-Noise Envelope Detector

The adaptive noise immunity module enables the demodulator to obtain adaptive noise immunity. Specifically, the charge pump initially demodulates the high frequency signal to obtain the initial demodulated signal. Then, it is output to the INP terminal of the comparator via the INP terminal. At the same time, the demodulated signal is again passed through the envelope detector composed of NM0 and C0 devices and output to the INM terminal of the comparator via the INM terminal. In particular, the output signal at the INM terminal is the envelope signal of the output signal at the INP terminal. Figure 7 shows the above structure and the body terminal of NM0 is connected to the source terminal [15].

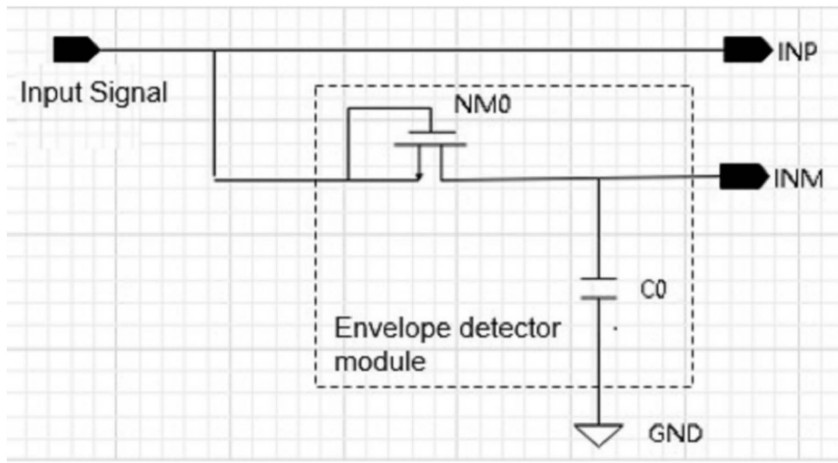

**Figure 7.** Envelope detector.

The output signal at the INM end is slightly delayed in phase compared to the signal at the INP end because it is re-checked by the envelope detector. In this paper, we use the ASK modulated signal as the output signal to the demodulator, and then measure the output signal at INM and INP, as shown in Figure 8. The peaks and valleys of the output signal at the INM terminal are 1 to 2 μs slower than the output signal at the INP terminal, but the peak values and waveforms of the two are similar. In other words, the comparator can perform a differential comparison of the hysteresis characteristics of the output signals at the INM and INP ends, i.e., adaptive noise immunity. Therefore, the demodulator of this design has good noise immunity performance.

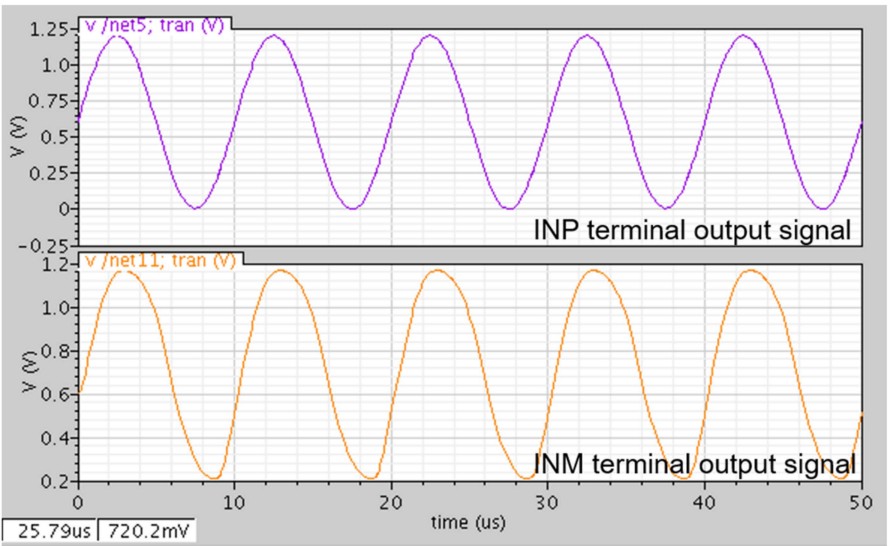

**Figure 8.** Comparison of delay difference between INP and INM.

### 2.2.3. Hysteresis Input Comparator with High Common Mode Input and Dual Differential Op-Amps

The comparator consists of a bias circuit and a differential op-amp. The functions are as follows. The bias circuit is designed so that the demodulator has a high common-mode input range and low power consumption. Differential operations are performed on the INM and INP signals to further demodulate the signals to obtain the final square wave, and the effects from the internal demodulator are weakened due to operating temperature rise.

The bias circuit provides a stable current for the differential op-amp and current mirror sections and a suitable operating area for the differential op-amp section. Details of the comparator operation process are described below. Figure 9 shows the differential op-amp module intercepted in the comparator.

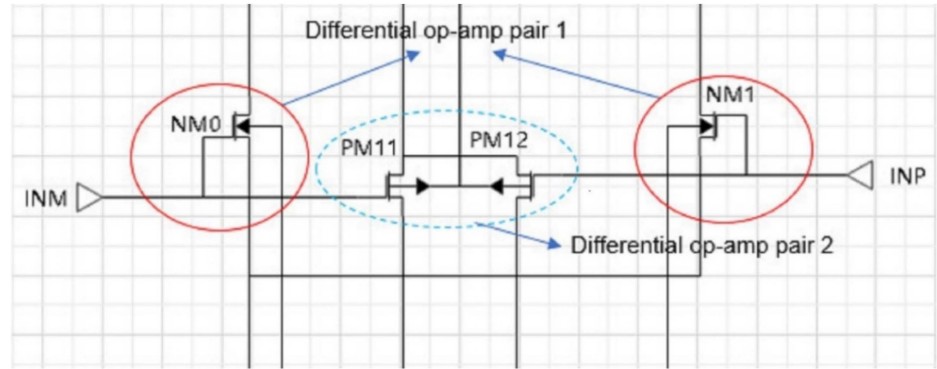

**Figure 9.** Differential amplifier.

The following section briefly introduces the comparator operation process in combination with Figure 10, taking as an example when the INM voltage value is higher than a certain value of INP. For the output affected by INP, the NM1 drain voltage directly reaches the maximum voltage of 1.2 V due to the amplification of approximately a thousand times, and the PM5 gate voltage is the NM1 drain voltage, and PM5 does not conduct. Since the PM12 drain voltage is about 0.3 V, the NM11 gate is high potential |VGS|>|Vth|, NM11 conducts and the output at the left-OUT terminal is 0 V. For the output of INM influence, NM0 drain voltage is 0.85 V, PM0 gate potential is lower than the source potential, |VGS|>|Vth|, PM0 conducts, thus pulling up NM2, NM5 gate voltage, thus making the two NMOS conductive. Since PM11 drain voltage is 0 V and NM6 gate voltage is 0 V, NM6 does not conduct, making PM2, PM7 gate at high potential, thus PM2, PM7 does not conduct and the right OUT terminal output is 0 V.

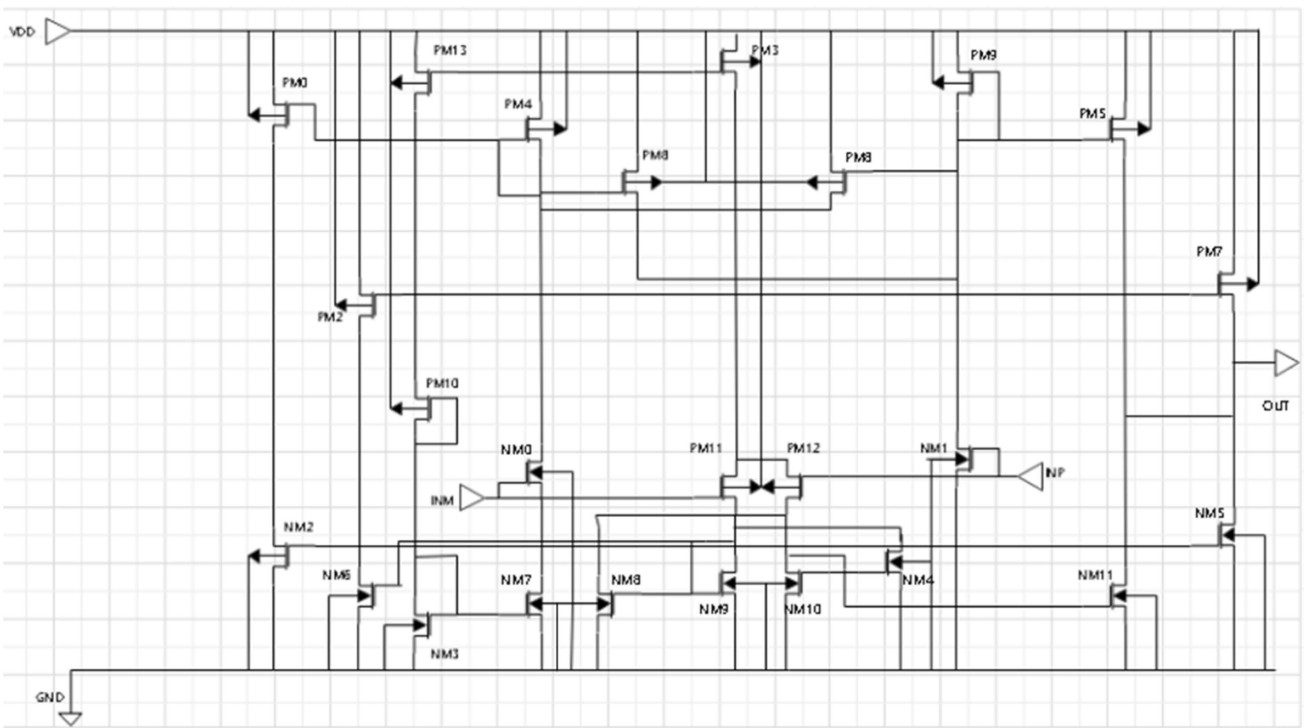

**Figure 10.** Comparator.

The reference voltage of the demodulator is 1.2 V. After the input signal is differentially compared and demodulated by the comparator, the output signal waveform approximates a square wave so that a simple inverter can be used. The complex Schmitt trigger can be discarded for shaping, thus saving chip area. When the difference between INM and INP input voltage is lower than 0.02 V, the output voltage starts to have obvious distortion. The output voltage is no longer a standard square wave, which only produces partial distortion at the signal intersection for the hysteresis input signal pair of this design, which does not affect the overall signal demodulation. The simulation shown in Figure 11 is obtained using INM fixed value and INP single incremental input to obtain the voltage variation at the output of two differential op-amps composed of PMOS and NMOS, respectively.

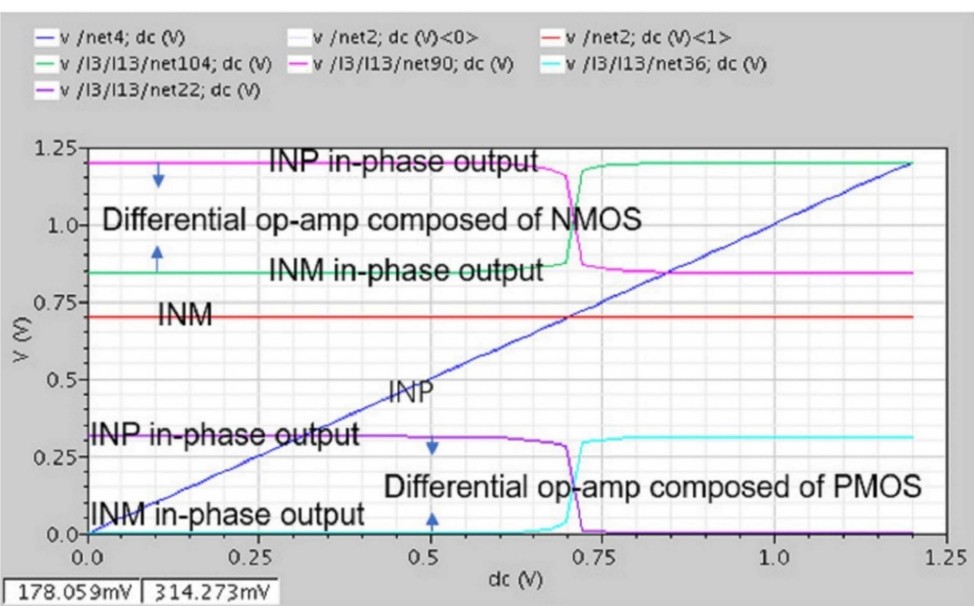

**Figure 11.** Differential amplifier IN/PUT.

### 3. Results and Discussion

#### 3.1. Adaptive High Noise Immunity Performance Verification

The demodulator simulation is based on a 0.13 μm CMOS process. The reference voltage is 1.2 V. In this paper, the baseband signal is loaded onto the carrier to obtain the High Frequency Signal, and then a single incremental signal is superimposed for simulation to observe the noise distortion threshold of demodulation and verify the anti-noise performance of this demodulator. The simulated modulation is achieved by modulating a 100 KHz baseband signal to a 900 MHz UHF signal. Figure 12 is used to demonstrate the demodulation details:

1. The INP signal corresponds to the wave peak of the high-frequency signal envelope, verifying the correctness of the initial demodulation of the charge pump.
2. The INM signal is equivalent to the phase delay signal of the INP signal, i.e., the hysteresis signal of the INP signal, when only the peak and trough changes are concerned.
3. The wave peak of the high-frequency signal envelope is the wave peak of the baseband signal. The output signal corresponds to the wave peak of the high-frequency signal envelope, which verifies the correctness of the demodulation.

That is, Figure 12 verifies the correctness of the demodulator demodulation. In this figure, the top green signal is the 900 MHz ASK modulated signal, which is used as the input signal to the demodulator. The second pink signal is the output signal measured at the INP. The third blue signal is the output signal measured at INM. The fourth purple signal is the output signal measured at the comparator output. The last orange signal is the noise signal, which is used to measure the limit voltage of the demodulator to withstand noise.

This simulation verifies that the demodulator only starts to show severe distortion when the noise strength reaches around 3.729 V, which causes the comparator to not work properly. As can be seen in Figure 13, the high-frequency signal increases with a single incremental noise signal. At the same time, the low-frequency signal measured at the INP end gradually shifts up and gradually loses its waveform. Similarly, the envelope signal measured at the INM end gradually shifts upward and loses its waveform. This also causes the square wave measured at the demodulator output to change from an alternating 0/1 square wave to a 0 signal at some point. By comparing the output signal with the single

incremental noise signal, we can find that the demodulator does not work properly when the single incremental noise signal reaches 3.729 V.

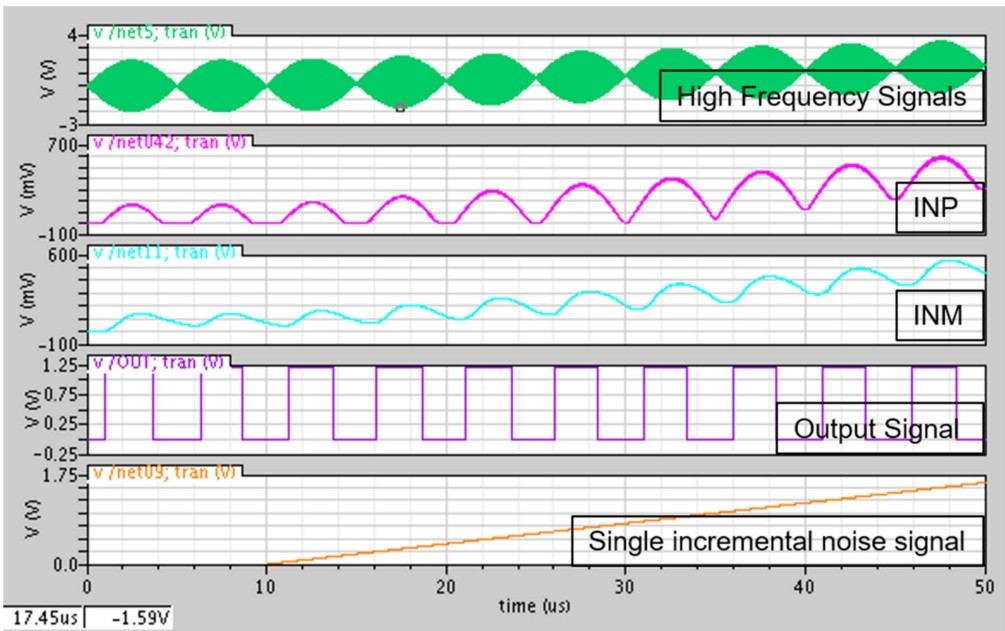

**Figure 12.** Verification of demodulation simulation correctness.

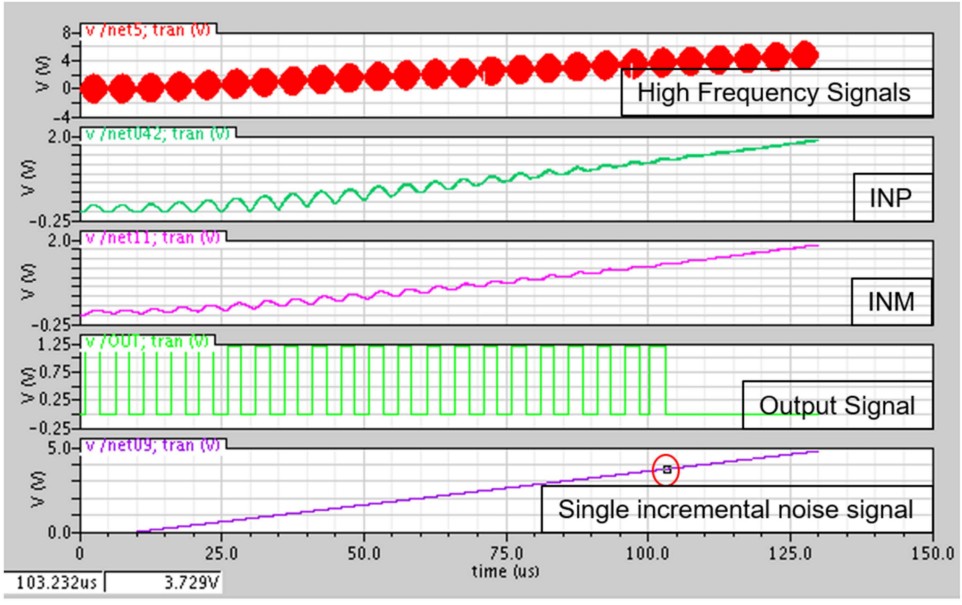

**Figure 13.** Simulation performance verification.

Additionally, in order to demonstrate the effect of temperature on this design, six temperature points were selected for simulation in the temperature range from 30 °C to 90 °C. Finally, the output waveform variations at different temperatures are measured at the output of the demodulator. As can be seen from Figure 14, when the temperature is between 33 °C and 78 °C, the output waveform does not change much and the temperature effect is negligible. However, when the temperature reaches 88.9 °C, the output waveform becomes a straight line and is completely distorted. In other words, this demodulator is designed to support demodulation up to at least 78 °C, which is suitable for daily life scenarios.

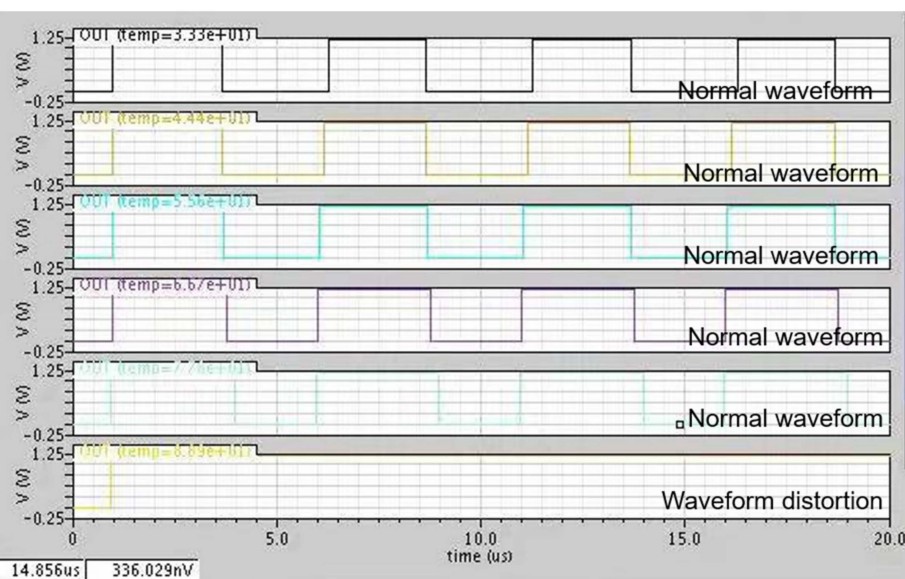

**Figure 14.** Comparison of output at different temperatures.

### 3.2. Low Power Verification

By comparing with repotted data in the references, it can be seen that the average power consumption of this design is 1/3000 of [4], 1/167 of [5], 1/294 of [6], and 1/3 of [7], and the average power consumption level of this demodulator is much lower than ones reported in the references. For passive UHF RFID, the ultra-low-power demodulator offers significant cost savings, high reliability and high noise immunity, making the demodulator more suitable for today's smart home and IoT detection. Figure 15 shows the waveform variations of INM input current, INP input current, and total current during the actual operation of the demodulator. The current peak size at the INM side is 197.43 nA, and the current peak at the INP side is 334.3 nA. The average power consumption is 113.6 nW, which is much lower than the micron-level current of references [4–6]. The average operating current of this design is 133.9 nA, which is lower than the 300 nA claimed by reference [7].

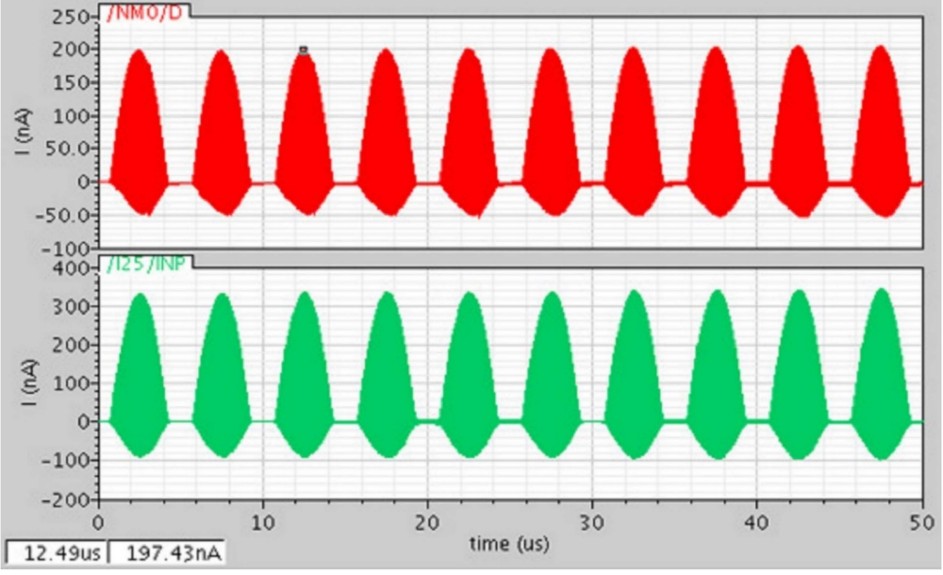

**Figure 15.** Electric current.

As can be seen from Table 1, the demodulator proposed in this paper uses a higher and wider transmission frequency with a carrier frequency approximately 60 times higher than the one claimed by references [4–6]. The demodulator enables information transmission over long distances with lower power consumption than low frequency carrier designs. Table 1 informatively lists the data comparison between this design and those reported in the references. Table 2 gives the technical specifications of different international ASK demodulators.

**Table 1.** Data comparison.

| Reference | Year | Craftsmanship (μm) | Carrier Frequency (MHz) | Data Rate (Kbps) | Average Power (μW) |
|-----------|------|--------------------|-------------------------|------------------|--------------------|
| In [4] | 2010 | 0.35 | 13.56 | 1228.8 | 306 |
| In [5] | 2016 | 0.18 | 5 | 500 | 17 |
| In [6] | 2019 | 0.18 | 13.56 | 1024 | 30 |
| In [7] | 2020 | 0.13 | 880–940 | - | 0.3 (nA) |
| This paper | 2021 | 0.13 | 900 | 128 | 0.114 |

**Table 2.** Technical indicators.

| Frequency | 125 KHz | 13.56 MHz | 900 MHz | 2.45 GHz |
|-----------|---------|-----------|---------|----------|
| Scope | 0.2 m | 0.5 m | 2–5 m | 1–2 m |
| Transmission rate | Less than 1 kbit/s | 25 kbit/s | 30 kbit/s | Higher than 100 kbit/s |
| Standard | ISO 18000-2 | ISO 18000-3 | ISO 18000-6 | ISO 18000-4 |

It is worth mentioning that in Table 1, the average current is used to characterize the power consumption since the specific voltage values and the specific power consumption values of the demodulator are not given in reference [7].

## 4. Conclusions

In this paper, a new demodulator design scheme based on a 0.13 μm COMS process is proposed. Its power consumption value is about one hundredth of that published in references [4–6] and about one third of that published in reference [7], and its power consumption is extremely low—approximately 114 nW. In terms of noise immunity, this demodulator can typically demodulate noise with voltages below 3.729 V. The absolute threshold value of the charge pump MOS tube is reduced from 381.742 mV to 303.397 mV. and the design can operate normally between 0 and 78 °C.

**Author Contributions:** N.B. and S.Y. designed the method and wrote the paper; H.L., J.-M.L. and Y.-H.X. performed the experiments and analyzed the data. All authors have read and agreed to the published version of the manuscript.

**Funding:** This research was funded by the National Natural Science Foundation of China (No.61204039) and the Key Laboratory of Computational Intelligence and Signal Processing, Ministry of Education (No. 2020A012).

**Institutional Review Board Statement:** Not applicable.

**Informed Consent Statement:** Not applicable.

**Data Availability Statement:** All data included in this study are available upon request by contacting with the corresponding author.

**Conflicts of Interest:** The authors declare that there are no conflict of interest regarding the publication of this paper.

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
