# Peer review of "Adaptive Noise-Resistant Low-Power ASK Demodulator Design in UHF RFID Chips"

_electronics, doi:10.3390/electronics10243168_

Round 1
Reviewer 1 Report
The abstract does not give the reader a clear idea of how your peoposed solution prevails over the traditional solutions. You need to compare your solution to the existing ones.
In the conclusion, you need to use the simulation results to prove that your proposed solution prevails over the others in market.
Author Response
Response to Reviewer 1 Comments
Point 1: The abstract does not give the reader a clear idea of how your peoposed solution prevails over the traditional solutions. You need to compare your solution to the existing ones.
Response 1: Thank you very much for your support and approval. Likewise, I apologize for taking so long to respond to your valuable revisions. The following is my revised summary.
This paper presents a novel signal demodulator for ultra-high frequency (UHF) radio frequency identification (RFID) tag chips. It has the advantages of high noise immunity, large input range and low power consumption. The antenna receives the amplitude shift keying (ASK) modulated signal from the reader, and then the demodulator demodulates the modulated signal to get the baseband signal. Finally, the digital signal obtained by demodulation is transmitted to the digital circuit and processed in the next step. The demodulator consists of a charge pump, an envelope detector, and a comparator. In particular, the demodulator provides a hysteresis input signal to the comparator through two envelope detectors, which results in better noise immunity. The demodulator is based on a 0.13 um standard CMOS process. In comparison, at low process level, the design is suitable for demodulating high frequency signals at 900 MHz, supporting data rates of 128 kbps. Simulations show that the demodulator consumes an average power as low as 113.6 nW and can demodulate normally within a noise voltage range of 1.5 times the peak of the original signal. Compared with the references, the demodulator proposed in this paper has better performance in terms of low power consumption and noise immunity.
Point 2: In the conclusion, you need to use the simulation results to prove that your proposed solution prevails over the others in market.
Response 2: By drawing on the way the references are written, I have made certain modifications to the conclusion section.
- The excellent aspects of the simulation results are highlighted.
- The less important descriptive statements that have been removed make the conclusion look more concise. The following is my revised conclusion.
In this paper, a new demodulator design scheme is proposed. The power consumption value is about one hundredth of the power consumption value declared in reference [4][5][6] and about 1/3 of the power consumption value declared in reference [7], with an extremely low power consumption of about 114 nW. In terms of noise immunity, the demodulator can normally demodulate noise with a voltage lower than 3.729 v. The demodulator has a good noise immunity performance.
Reviewer 2 Report
This paper mainly discusses the challenges of designing a low power ultra-high frequency (UHF) RFID demodulator. The obtained performances are interesting, however the paper needs to improve the methods and results section.
The reviewer has the following remarks:
- The Introduction section should be rewritten, as many of the sentences start with a citation. The sentences should be reformulated as a conclusion, or a short survey of the results presented in referenced papers.
- Every abbreviation should be fully explained before it is used: MI, ASK, RFID etc.
- The advantages enumerated in introduction section (lines 39 to 44) should be organized as a list (bullet list) using the MDPI recommended style.
- MDPI guidelines for equations writing should be followed. The text following an equation should be under the equation, and not as a new paragraph.
- Each equation should be introduced by a text explaining its purpose. Equation 4 and 5 should be rewritten.
- A sentence should not start with Figure x, where x is the figure number. A figure cross-reference should be introduced through a phrase.
- Figure 3 should be moved, so that is closer to the text.
- The connection between bias body, drain and gate should be detailed much more, to present the background decision (line 136):
- how much the threshold voltage was improved compared to bias body-source connection,
- what is the leakage current improvement,
- how is the temperature impact improved
- what is the effect from reliability point of view.
- The CMOS transistors graphical representation should be consistent (Figure 6). If the bias body pin is connected to source, then it should be stated.
- In Figure 9 it is not clear if the bias body pins of NM0, NM7 and NM8 transistors are connected to ground. Can you please mark the connection if that is the case.
- The sentences from lines 144 – 152 should be rephrased. Signals INM and INP are depicted as output signals in Figure 6, however in the text are referenced as input. It might help to mention that the obtained signals are used as input for the inverted (INP) and non-inverted input (INM) of the differential op-amp.
- The frequency units should be written according to International System of Units, kHz and MHz instead of KHZ respectively MHZ.
- Table 2 should be removed as it does not bring new information compared to Table 1.
- More details about the results presented in Figure 7 should be provided. For example, applying an ASK modulated signal and measuring the output signal at INM and INP. To have the output signal represented as a square wave signal (Figure 11), the input signal should have been like Figure 8 from reference [4]. The Results and Discussion section needs improvements, for example input and intermediary signal waveforms to prove the concept performances.
Author Response
Response to Reviewer 2 Comments
Point 1: The Introduction section should be rewritten, as many of the sentences start with a citation. The sentences should be reformulated as a conclusion, or a short survey of the results presented in referenced papers.
Response 1: Thank you very much for your valuable comments on my article. I am also very sorry that I did not reply to you in time. Here are the changes I made to the introduction section (the number of lines marked in my reply may be a little different from the number of lines in your suggestion, as I changed some images and drastically cut some paragraphs).
- Line 25. “The demodulator is used to reduce the high frequency modulated signal to a low frequency baseband signal.”
- Lines 29 to 39. “In [4], the authors forgo the use of large capacitors in their design in exchange for a small chip area and a good implantable chip experience, yet suffer from design complexity and high-power consumption. In [5], the authors claim that their circuit can operate with a minimum modulation index (MI) of 5.26% and 0.5 Mb/s by using shifter, switches and therefore can be well used in biomedical implants, but is not suitable for UHF applications due to the low switching speed. In [6], the authors used a hysteresis input structure design to reduce the noise impact and a secondary amplifier to increase the input signal range, but it still has the disadvantage of high-power consumption. In [7], the authors improve the efficiency of the voltage conversion from RF port to DC interface (RF-DC) by threshold voltage compensation, leakage current suppression and dynamic voltage adjustment schemes, but consume a larger chip area.”
- Line 40. “To overcome mentioned disadvantages,”
Point 2: Every abbreviation should be fully explained before it is used: MI, ASK, RFID etc.
Response 2: I added specific explanations when each abbreviation was first used.
- Line 8. “ultra-high frequency (UHF)”
- Line 9. “radio frequency identification (RFID)”
- Line 10. “amplitude shift keying (ASK)”
- Line 32. “modulation index (MI)”
- Line 79. “ is the equivalent omnidirectional radiated power”
Point 3 The advantages enumerated in introduction section (lines 39 to 44) should be organized as a list (bullet list) using the MDPI recommended style.
Response 3: I have revised the format of the introduction and made certain changes to other sections.
Lines 43 to 48. “
- It has the advantage of adaptive noise reduction and can still work effectively under high noise conditions.
- The comparator section, designed with two differential op-amps, shares a common bias circuit, and the outputs are directly coupled to reduce demodulation errors caused by internal circuit heat generation.
- Compared with the data given in the references, the power consumption value of this design is about one hundredth of the power consumption value given in the references [4][5][6] and the power consumption value of this design is about 1/3 of the power consumption value given in the reference [7]. This paper gives specific data comparisons in Section 3.2. [8,9,10].”
Lines 229 to 234. “
- The INP signal corresponds to the wave peak of the high-frequency signal envelope, verifying the correctness of the initial demodulation of the charge pump.
- The INM signal is equivalent to the phase delay signal of the INP signal, i.e., the hysteresis signal of the INP signal, when only the peak and trough changes are concerned.
- The wave peak of the high-frequency signal envelope is the wave peak of the baseband signal. The output signal corresponds to the wave peak of the high-frequency signal envelope, which verifies the correctness of the demodulation.”
Point 4: MDPI guidelines for equations writing should be followed. The text following an equation should be under the equation, and not as a new paragraph.
Response 4: By referring to the MDPI format, I have modified the formula section so that the formula is introduced in a complete sentence with the variables below.
Lines 76 to 97. “
The reading distance of the reader, i.e., the working ranger of the reader and RFID tag, can be derived from Equation 1:
(1)
while is the mismatch factor, is the equivalent omnidirectional radiated power, is the polarization loss factor, is the code piece threshold power, is the tag antenna gain, and is the free-space wavelength. The mismatch factor and polarization loss factor can be 0.5, can be obtained from Equation 2. The tag antenna gain can be obtained from Equation 3. , is the speed of light, and is the operating frequency. The code piece threshold power determines the lower power at which the tag works. namely, the tag can work properly only when the input power exceeds . The formulas for and are given below:
(2)
(3)
while is the transmit power, is the antenna gain, is the loss on the feed line, is the free space wavelength, and is the effective area of the antenna.
For ASK demodulators, the envelope detector is an important component to achieve its demodulation function [13,14]. The following is the rationale for the envelope detector to demodulate the baseband signal from the UHF signal (860-960 MHz). The envelope detector model is shown in Figure 3. The RC limit formula is given below:
(4)
while is the subsequent circuit resistance, is angular velocity of the local carrier, is the capacitance capacity, and is the baseband signal frequency.”
Point 5: Each equation should be introduced by a text explaining its purpose. Equation 4 and 5 should be rewritten.
Response 5: For Equation 4 and Equation 5, I partially modified the formula and text content to make them correspond to each other.
Lines 94 to 107. “
The envelope detector model is shown in Figure 3. The RC limit formula is given below:
(4)
while is the subsequent circuit resistance, is angular velocity of the local carrier, is the capacitance capacity, and is the baseband signal frequency.
Figure 3. Working model of the envelope detector.
For the envelope detector, improper settings of the resistor R and capacitor C can produce inert distortion. Specifically, if the product of C values is too large, it will result in too large a frequency gap between charge and discharge, thus making the output signal change too slowly. To prevent this phenomenon, the limiting equation is given below:
(5)
while detector subsequent circuit total resistance value, and refers to the capacitor resistance, refers to the input signal frequency, and the for the amplitude modulation coefficient. usually takes the value 1.5.”
Point 6: A sentence should not start with Figure x, where x is the figure number. A figure cross-reference should be introduced through a phrase.
Response 6: For the numbering of figures appearing in the text, I have used a cross-reference format.
Line 60. “The flow chart shown in Figure 1”
Line 69. “as shown in Figure 2”
Line 94. “is shown in Figure 3”
Line 133. “is shown in Figure 4.”
Line 147. “as shown in Figure 5” “Figure 6 shows the results of”
Line 169. “Figure 7 shows the above”
Line 177. “as shown in Figure 8”
Line 192. “Figure 9 shows the differential op-amp”
Line 197. “process in combination with Figure 10”
Line 217. “The simulation shown in Figure 11 is”
Line 228. “Figure 12 is used to”
Line 238. “Figure 12 verifies the correctness of”
Line 249. “As can be seen in Figure 13”
Line 268. “As can be seen from Figure 14”
Line 282. “Figure 15 shows the waveform”
Line 289. “As can be seen from Table 1”
Line 293. “Table 1 informatively lists”
Line 294. “Table 2 gives the technical specifications of”
Point 7: Figure 3 should be moved, so that is closer to the text.
Response 7: I have moved the position of Figure 3.
Lines 91 to 99. “
For ASK demodulators, the envelope detector is an important component to achieve its demodulation function [13,14]. The following is the rationale for the envelope detector to demodulate the baseband signal from the UHF signal (860-960 MHz). The envelope detector model is shown in Figure 3. The RC limit formula is given below:
(4)
while is the subsequent circuit resistance, is angular velocity of the local carrier, is the capacitance capacity, and is the baseband signal frequency.
Figure 3. Working model of the envelope detector.”
Point 8: The connection between bias body, drain and gate should be detailed much more, to present the background decision (line 136):
- how much the threshold voltage was improved compared to bias body-source connection,
- what is the leakage current improvement,
- how is the temperature impact improved
- what is the effect from reliability point of view.
Response 8: I have answered the first three questions. But since I don't know much about using cadence, I can't give a specific simulation result for reliability.
- Lines 155 to 158. “In this design, when the body terminal is connected to the source, the threshold voltage is -381.742 mV. However, when the body terminal is connected to the drain and gate, the threshold voltage becomes -303.397 mV”
- I designed the charge pump considering the case where the input signal is alternately positive and negative. In other words, the part of the input signal greater than zero is filtered, while for the part of the input signal less than zero, it flows out through the PM0 tube. And in the latter part of the article according to the advice of my mentor used the signal with non-negative values, so it seems that this sentence is abrupt. I have revised the text of the paragraph.
- Lines 265 to 275. I added the simulation of the temperature. “
Also, in order to demonstrate the effect of temperature on this design, six temperature points were selected for simulation in the temperature range from 30 degrees Celsius to 90 degrees Celsius. Finally, the output waveform variations at different temperatures are measured at the output of the demodulator. As can be seen from Figure 14, when the temperature is between 33 degrees C and 78 degrees C, the output waveform does not change much and the temperature effect is negligible. However, when the temperature reaches 88.9 degrees Celsius, the output waveform becomes a straight line and is completely distorted. In other words, this demodulator is designed to support demodulation up to at least 78 degrees Celsius, which is suitable for daily life scenarios.
Figure 14 Comparison of output at different temperatures”
Point 9: The CMOS transistors graphical representation should be consistent (Figure 6). If the bias body pin is connected to source, then it should be stated.
Response 9: I have revised the content of the article.
Line 170. I have added some text to the article to make this connection more clear to the reader. “Figure 7 shows the above structure and the body terminal of NM0 is connected to the source terminal.”
Point 10: In Figure 9 it is not clear if the bias body pins of NM0, NM7 and NM8 transistors are connected to ground. Can you please mark the connection if that is the case.
Response 10: I am very sorry, because of my work error, i.e. I missed a line, which led to the confusion of the ground wire at the body end of NM0, NM7 and NM8, I have re-filled it.
Line 208. The red part of the picture below is the ground wire I added to the NM7 and NM8 body ends.
Point 11: The sentences from lines 144 – 152 should be rephrased. Signals INM and INP are depicted as output signals in Figure 6, however in the text are referenced as input. It might help to mention that the obtained signals are used as input for the inverted (INP) and non-inverted input (INM) of the differential op-amp.
Response 11: I am sorry for the confusion I caused you by my poor choice of words. I have corrected the word in the text.
Lines 174 to 176. “The output signal at the INM end is slightly delayed in phase compared to the signal at the INP end because it is re-checked by the envelope detector. In this paper, we use the ASK modulated signal as the output signal to the demodulator, and then measure the output signal at INM and INP, as shown in Figure 8.”
Point 12: The frequency units should be written according to International System of Units, kHz and MHz instead of KHZ respectively MHZ.
Response 12: I have made changes in the article. And corrected the error of using lowercase v for voltage in some parts.
Line 17. “900 MHz”, “128 Kbps”
Line 93. “860-960 MHz”
Line 199. “1.2 V”
Line 201. “0.3 V”
Line 202. “0 V”
Line 203. “0.85 V”
Line 207. “0 V”
Line 227. “100 KHz”, “900 MHz”
Line 296. “MHz”, “Kbps”
Line 298. “125 KHz”, “13.56 MHz”, “900 MHz”, “2.45 GHz”
Point 13: Table 2 should be removed as it does not bring new information compared to Table 1.
Response 13: I have deleted table 2.
Line 298.
Point 14: More details about the results presented in Figure 7 should be provided. For example, applying an ASK modulated signal and measuring the output signal at INM and INP. To have the output signal represented as a square wave signal (Figure 11), the input signal should have been like Figure 8 from reference [4]. The Results and Discussion section needs improvements, for example input and intermediary signal waveforms to prove the concept performances.
Response 14: Thank you very much for your suggestion, it solved my confusion because I couldn't describe a picture very well. Here is what I have modified. And I have added some notes to Figure 13.
- Lines 246 to 263. “This simulation verifies that the demodulator only starts to show severe distortion when the noise strength reaches around 3.729v, and this causes the comparator not to work properly. As can be seen in Figure 13, as the single incremental noise signal (in order from top to bottom, this is the fifth signal, which is the purple one. The high frequency signal (this is the first signal, which is the red signal) is gradually shifted upward as the single incremental noise signal (in order from top to bottom, this is the fifth signal, which is the purple signal. This is also described later in this order) increases. At the same time, the low-frequency signal measured at the INP end (this is the second signal, which is the green signal) gradually shifts upward and gradually loses its waveform. Similarly, the envelope signal measured at the INM end (this is the third signal, the pink signal) is also gradually shifted upward and loses its waveform. This also causes the square wave measured at the output of the demodulator (this is the fourth signal, which is the square wave) to change its output from a 0/1 alternating square wave to a 0 signal at some point. By comparing the output signal with the single incremental noise signal, we can find that the demodulator does not work properly when the single incremental noise signal reaches 3.729V.
Figure 13. Simulation performance verification.”
- However, I am sorry to say that the waveform in Figure 8 of reference [4] did not appear in my transient simulation, although I set the parameter "transient noise". The figure below is the waveform I obtained, and since I am not sure of the correctness of the figure, I did not add it to the article, but just show you the picture.
Reviewer 3 Report
MDPI Electronics - full paper
Adaptive Noise-Resistant Low-Power ASK Demodulator Design in UHF RFID Chips
Sent on 24-Nov 2021 at 01 :30
- General comments
In this work authors claim the design of an ASK demodulator, oriented to EFID chip applications at 900 MHz, of low power consumption and high resistance to noise and wide input range. This demodulator consists of a low-threshold charge pump, an adaptive noise-resistant enve-256 lope detector, and a hysteresis input comparator as reported in this paper. As shown by simulations of figure 12, the designed demodulator can work normally for a noise amplitude of 0-3.729 V which is about 1.5 times of the high-frequency signal amplitude. Average consumption power of this design is about 0.114 uW which makes it very economic compared to other designs reported in references mentioned by author.
However, author has to care about writing style, especially, at the introduction to encourage reader to continue reading its paper. Use short and meaningful sentences, use conjunctions that join words which link together parts of a sentence (for, so, that why, and, but, or…) and use verbs to clarify your idea. I think author has to look for an English teacher to help him reorganize and rewrite some parts of his work. Also figures have to be set at the middle.
Abbreviations: the scientific community uses commonly a lot of abbreviations to stand for scientific, algorithmic or technological concepts, so any given acronyms have to be explained such as EIRP.
In overall this paper is well organized, comparison results with other works are presented. This work deserves to be published with major revision as given in details below in order to enhance the quality of this paper.
- Detailed comments
Introduction
In the introduction, author begins directly by a reference number, he should say, for example, “ according to …” . “[4] circuit design does not….”, “[5] operates at 5.26% MI and 0.5 Mb/s b…..” and “[7] uses threshold compensation…”
At the end of the introduction author should talk about the organization of his paper. For example: “This paper is organized as follow…” to make reader understand the manuscript structure easily.
This phrase has to be rearranged: “Demodulator that is, through digital signal processing technology, the modulation of the obtained high-frequency signal is reduced to the low-frequency information signal.”
“[4] circuit design does not use large capacitors, and the advantage is the chip area is 27 small, the disadvantage is that the design is complex” ??? we can’t begin a phrase by a reference number. Author has to use some conjunctions like “so advantages..”, “but disadvantages …”
“[5] operates at 5.26% MI and 0.5 Mb/s by using shifters…”. Who does operate at 5.26% ?? [5]??
“[6] reference hysteresis structure to suppress noise…”. Does author mean reference [6] presents a hysteresis structure so suppress noise? Sentence has to be reformulate.
This sentence has to be revised please: “the choice of using a secondary amplifier in increasing the input signal range, and the use of a certain number of resistor devices, so that the design has the advantages of high noise immunity and low modulation index”. It needs a verb; “the choice of…..verb….”
“[7] uses threshold compensation…”. Suggestion: “In [7] authors used ….”
This beginning of the 3rd paragraph is not suitable: “Compared with the above problems,…”. Efforts of other peoples are not “problems”!!. Author, for example, could say: “To overcome mentioned disadvantages, …”
“(3) The power consumption of the proposed design is about 1/3 of that of [7] and about one percent of that of other references compared to the power consumption of each reference, and detailed data comparison is given in the low-power module (3.2) [8,9,10]”. Suggestion: “, and (3) the power…”, this part has to be rearranged: “and about one percent of that of other references compared to the power consumption of each reference”. “, and detailed data comparison is given in the low-power module (3.2) [8,9,10]”, suggestion: “. Detailed data comparison is given in the low-power module section (3.2) [8,9,10]”
Section Methods:
“…interfered with by noise.” Do author mean “…interfered with noise”?
“The code piece threshold power ?ℎ determines the lower frequency at which the tag works”. Correction: “The code piece threshold power ?ℎ determines the lower power at which the tag works”
In most of literature we find read range expressed as follow: , so in your case EIRP has to be equal to . If we consider the loss of feeding line, EIRP should be expressed as follow: EIRP=(Pt-Lc).Gt. Please try to revise this formula.
“…the angular velocity of the local carrier”, correction: “w the angular velocity of the local carrier”
“..distortion, that is, because the RC is too large”??
“..resulting in too large a gap between the charging and discharging frequencies” suggestion: “…too large gap…”
“The ? for the envelope detector…”, suggestion: “R refers to the envelope detector…”
“? for the capacitor resistance, the ? for the input signal frequency…”. Suggestion: “, and C refers to…, whereas ? refers to…”
Line 97: Please rearrange this sentence: “√1−??2/?? is generally desirable 1.5 is used for calculation.” ???
Line 104: “(3) for passive 104 RFID, RFID tags directly from the received RF signal to extract energy and supply other circuits”, where is the verb of “RFID tags”?
Line 106: “so the smaller the power consumption of the demodulator, the faster the response 106 of the tag.” What is the relationship between power consumption and speed of response? Please give a reference.
Section 2.2:
Line 112: “(1) the tuned signal is passed through the charge pump inside the charge pump..” through or inside the charge pump??
Section 2.2.1:
Line 138: could author explain more this: “and the PMOS with diode action forms a filter to filter out the reverse current”?
Line 210: full stop mark before “The output signal…”
Line 216: “This simulation verifies that the demodulated signal starts to show serious distortion only when the noise strength reaches around 3.729 V, which causes the comparator not to work properly.”. Author has to add legends to figure 12 to illustrate, for example, “noise signal”
Line 223: this sentence is repeated: “1.5 times the amplitude of the high-frequency signal in this simulation.”
Line 228: “By comparing with the data of the references”, suggestion: “By comparing with repotted data in the references”
Line 229: author has to preset the formula showing how to calculate the demodulator’ power consumption first, and corresponding references.
Line 230: “average power consumption level of this demodulator is much lower than that of the references”, suggestion: “average power consumption level of this demodulator is much lower than ones reported in the references”
Line 236: “The average power consumption is 133.9 nA and 113.6 nW”, do you mean 133.9 nW?
line 237: “References” not “reference”, line 243: “Table 1 informatively lists the data comparison between this design and the references.” Suggestion: “Table 1 informatively lists the data comparison between this design and the ones of references.”
Line 238: “the 300 nA power consumption of the demodulator in [7].” 300nW??
Line 239: “From the above table, it can be seen…”, which table? There is no table before this line!!
Lines 239-241: please insert table confirming the given information: “it can be seen that the present demodulator has a higher and wider transmission bandwidth with a carrier frequency up to 60 times higher than the low carrier frequency used in [1][2][3].”
Line 241: could author explain this idea: “The demodulator enables information transmission over long distances with lower power consumption than low frequency carrier designs” because from formula (1) the read range distance increases when wave length increases, this occurs when frequency decreases since . Does author mean a low power consumption design allows a longer read distance than a low frequency design?
Line 243: “Table 1 informatively lists the data comparison between this design and the references” suggestion: “Table 1 informatively lists the data comparison between this design and the ones reported in references”
Table 1: 0.3 nA or nW??
Line 249, Table1: replace “times” by “years” or “date”
Author Response
Response to Reviewer 3 Comments
Point 1: Abbreviations: the scientific community uses commonly a lot of abbreviations to stand for scientific, algorithmic or technological concepts, so any given acronyms have to be explained such as EIRP.
Response 1: Thank you very much for your approval of this article and the valuable comments you made. I have benefited greatly from these revisions. However, because I have changed a lot of text, it may make the number of lines in my response a little different from the number of lines you suggested. Here are the specific changes I made.
I added specific explanations when each abbreviation was first used.
- Line 8. “ultra-high frequency (UHF)”
- Line 9. “radio frequency identification (RFID)”
- Line 10. “amplitude shift keying (ASK)”
- Line 32. “modulation index (MI)”
- Line 79. “ is the equivalent omnidirectional radiated power”
Point 2: In the introduction, author begins directly by a reference number, he should say, for example, “ according to …” . “[4] circuit design does not….”, “[5] operates at 5.26% MI and 0.5 Mb/s b…..” and “[7] uses threshold compensation…”
Response 2: Here are my revisions to the introduction.
Lines 29 to 39. “In [4], the authors forgo the use of large capacitors in their design in exchange for a small chip area and a good implantable chip experience, yet suffer from design complexity and high-power consumption.3.7 In [5], the authors claim that their circuit can operate with a minimum modulation index (MI) of 5.26% and 0.5 Mb/s by using shifter, switches and therefore can be well used in biomedical implants, but is not suitable for UHF applications due to the low switching speed. 3.8 In [6], the authors used a hysteresis input structure design to reduce the noise impact and a secondary amplifier to increase the input signal range, but it still has the disadvantage of high-power consumption. In [7], the authors improve the efficiency of the voltage conversion from RF port to DC interface (RF-DC) by threshold voltage compensation, leakage current suppression and dynamic voltage adjustment schemes, but consume a larger chip area.”
Point 3: At the end of the introduction author should talk about the organization of his paper. For example: “This paper is organized as follow…” to make reader understand the manuscript structure easily.
Response 3: I have added a description of the structure of the article in the last part of the introduction.
Lines 52 to 55. “This paper is organized as follows. Section 2 describes the role of demodulator in the communication system, the general structure of UHF RFID, and the design details of each module of the demodulator. Section 3 discusses the simulation results of the demodulator in terms of noise immunity and low power consumption. Section 4 concludes the paper.”
Point 4: This phrase has to be rearranged: “Demodulator that is, through digital signal processing technology, the modulation of the obtained high-frequency signal is reduced to the low-frequency information signal.”
Response 4: I have reworked the statement to make it readable.
Line 25. “he demodulator is used to reduce the high frequency modulated signal to a low frequency baseband signal.”
Point 5: “[4] circuit design does not use large capacitors, and the advantage is the chip area is 27 small, the disadvantage is that the design is complex” ??? we can’t begin a phrase by a reference number. Author has to use some conjunctions like “so advantages..”, “but disadvantages …”
Response 5: I have reworked the statement to make it readable.
Lines 29 to 31. “In [4], the authors forgo the use of large capacitors in their design in exchange for a small chip area and a good implantable chip experience, yet suffer from design complexity and high-power consumption.”
Point 6: “[5] operates at 5.26% MI and 0.5 Mb/s by using shifters…”. Who does operate at 5.26% ?? [5]??
Response 6: I apologize for the oversight that prevented some words from being included in the text, and I have reworked them.
Lines 31 to 34. “In [5], the authors claim that their circuit can operate with a minimum modulation index (MI) of 5.26% and 0.5 Mb/s by using shifter, switches and therefore can be well used in biomedical implants, but is not suitable for UHF applications due to the low switching speed.”
Point 7: “[6] reference hysteresis structure to suppress noise…”. Does author mean reference [6] presents a hysteresis structure so suppress noise? Sentence has to be reformulate.
Response 7: I have reworked the statement to make it readable.
Lines 34 to 36. “In [6], the authors used a hysteresis input structure design to reduce the noise impact and a secondary amplifier to increase the input signal range, but it still has the disadvantage of high-power consumption.”
Point 8: This sentence has to be revised please: “the choice of using a secondary amplifier in increasing the input signal range, and the use of a certain number of resistor devices, so that the design has the advantages of high noise immunity and low modulation index”. It needs a verb; “the choice of….verb….”
Response 8: I rewrote the sentence.
Lines 34 to 36. “the authors used a hysteresis input structure design to reduce the noise impact and a secondary amplifier to increase the input signal range, but it still has the disadvantage of high-power consumption.”
Point 9: “[7] uses threshold compensation…”. Suggestion: “In [7] authors used ….”
Response 9: I have reworked the statement to make it readable.
Line 36. “In [7], the authors improve”
Point 10: This beginning of the 3rd paragraph is not suitable: “Compared with the above problems,…”. Efforts of other peoples are not “problems”!!. Author, for example, could say: “To overcome mentioned disadvantages, …”
Response 10: I am ashamed of my poor choice of words. I have made the correction.
Line 40. “To overcome mentioned disadvantages,”
Point 11: “(3) The power consumption of the proposed design is about 1/3 of that of [7] and about one percent of that of other references compared to the power consumption of each reference, and detailed data comparison is given in the low-power module (3.2) [8,9,10]”. Suggestion: “, and (3) the power…”, this part has to be rearranged: “and about one percent of that of other references compared to the power consumption of each reference”. “, and detailed data comparison is given in the low-power module (3.2) [8,9,10]”, suggestion: “. Detailed data comparison is given in the low-power module section (3.2) [8,9,10]”
Response 11: I made changes to the third sentence and modified the typography of these three sentences.
Lines 42 to 51. “
- It has the advantage of adaptive noise reduction and can still work effectively under high noise conditions;
- The comparator section, designed with two differential op-amps, shares a common bias circuit, and the outputs are directly coupled to reduce demodulation errors caused by internal circuit heat generation
- Compared with the data given in the references, the power consumption value of this design is about one hundredth of the power consumption value given in the references [4][5][6] and the power consumption value of this design is about 1/3 of the power consumption value given in the reference [7]. This paper gives specific data comparisons in Section 3.2.”
Point 12: “…interfered with by noise.” Do author mean “…interfered with noise”?
Response 12: I have reworked the statement to make it readable.
Line 59. “Since the signals received by RFID tags are transmitted in a wireless channel and are affected by noise, this issue needs to be considered when designing the demodulator.”
Point 13: “The code piece threshold power ?ℎ determines the lower frequency at which the tag works”. Correction: “The code piece threshold power ?ℎ determines the lower power at which the tag works”
Response 13: I have corrected the incorrect wording that appears in the text.
Line 85. “The code piece threshold power determines the lower power at which the tag works.”
Point 14: In most of literature we find read range expressed as follow: , so in your case EIRP has to be equal to . If we consider the loss of feeding line, EIRP should be expressed as follow: EIRP=(Pt-Lc)·Gt. Please try to revise this formula.
Response 14: I have modified this formula.
Line 87. “ ”
Point 15: “…the angular velocity of the local carrier”, correction: “w the angular velocity of the local carrier”
Response 15: I have fixed the error.
Line 96. “while is the subsequent circuit resistance, is angular velocity of the local carrier,”
Point 16: “..distortion, that is, because the RC is too large”??
Response 16: I have reworked the statement to make it readable.
Lines 100 to 103. “For the envelope detector, improper settings of the resistor R and capacitor C can produce inert distortion. Specifically, if the product of C values is too large, it will result in too large a frequency gap between charge and discharge, thus making the output signal change too slowly.”
Point 17: “..resulting in too large a gap between the charging and discharging frequencies” suggestion: “…too large gap…”
Response 17: I have reworked the statement to make it readable.
Line 102. “it will result in too large gap between charge and discharge”
Point 18: “The ? for the envelope detector…”, suggestion: “R refers to the envelope detector…”
Response 18: I have reworked the statement to make it readable.
Line 105. “while R refers to the envelope detector”
Point 19: “? for the capacitor resistance, the ? for the input signal frequency…”. Suggestion: “, and C refers to…, whereas ? refers to…”
Response 19: I have reworked the statement to make it readable.
Line 106. “and refers to the capacitor resistance, refers to the input signal frequency”
Point 20: Line 97: Please rearrange this sentence: “√1−??2/?? is generally desirable 1.5 is used for calculation.” ???
Response 20: I have reworked the statement to make it readable.
Line 107. “ usually takes the value 1.5.”
Point 21: Line 104: “(3) for passive 104 RFID, RFID tags directly from the received RF signal to extract energy and supply other circuits”, where is the verb of “RFID tags”?
Response 21: I have reworked the statement to make it readable.
Line 120. “the RFID tag functions by extracting energy from the received RF signal and thus for other circuits.”
Point 22: Line 106: “so the smaller the power consumption of the demodulator, the faster the response 106 of the tag.” What is the relationship between power consumption and speed of response? Please give a reference.
Response 22: The text is taken from Baidu.com, but I have quoted it without verifying it, for which I am very sorry. Since I could not verify the conclusion, I have removed it from the text.
Point 23: Line 112: “(1) the tuned signal is passed through the charge pump inside the charge pump..” through or inside the charge pump??
Response 23: I have reworked the statement to make it readable.
Line 127. “The tuned signal is initially demodulated by a charge pump in order to demodulate the baseband signal from the 900MHz UHF carrier signal.”
Point 24: Line 138: could author explain more this: “and the PMOS with diode action forms a filter to filter out the reverse current”?
Response 24: I designed the charge pump considering the case where the input signal is alternately positive and negative. In other words, the part of the input signal greater than zero is filtered, while for the part of the input signal less than zero, it flows out through the PM0 tube. And in the latter part of the article according to the advice of my mentor used the signal with non-negative values, so it seems that this sentence is abrupt. I have deleted this text.
Point 25: Line 210: full stop mark before “The output signal…”
Response 25: I have reworked the statement to make it readable.
Line 235. “signal. The output”
Point 26: Line 216: “This simulation verifies that the demodulated signal starts to show serious distortion only when the noise strength reaches around 3.729 V, which causes the comparator not to work properly.”. Author has to add legends to figure 12 to illustrate, for example, “noise signal”
Response 26: Thank you for your comments to help me understand how to describe a result. The following is my revised version.
Lines 246 to 261. “This simulation verifies that the demodulator only starts to show severe distortion when the noise strength reaches around 3.729v, and this causes the comparator not to work properly. As can be seen in Figure 13, as the single incremental noise signal (in order from top to bottom, this is the fifth signal, which is the purple one. The high frequency signal (this is the first signal, which is the red signal) is gradually shifted upward as the single incremental noise signal (in order from top to bottom, this is the fifth signal, which is the purple signal. This is also described later in this order) increases. At the same time, the low-frequency signal measured at the INP end (this is the second signal, which is the green signal) gradually shifts upward and gradually loses its waveform. Similarly, the envelope signal measured at the INM end (this is the third signal, the pink signal) is also gradually shifted upward and loses its waveform. This also causes the square wave measured at the output of the demodulator (this is the fourth signal, which is the square wave) to change its output from a 0/1 alternating square wave to a 0 signal at some point. By comparing the output signal with the single incremental noise signal, we can find that the demodulator does not work properly when the single incremental noise signal reaches 3.729V.”
Point 27: Line 223: this sentence is repeated: “1.5 times the amplitude of the high-frequency signal in this simulation.”
Response 27: I am very sorry for my negligence. I have corrected it in the article.
Line 261.
Point 28: Line 228: “By comparing with the data of the references”, suggestion: “By comparing with repotted data in the references”
Response 28: I have reworked the statement to make it readable.
Line 276. “By comparing with repotted data in the references,”
Point 29: Line 229: author has to preset the formula showing how to calculate the demodulator’ power consumption first, and corresponding references.
Response 29: I add the power consumption calculation formula in Section 2.1.
Lines 108 to 115. “At the end, a simple power consumption calculation formula is given below to meet the calculation requirements in this paper. The formula is shown as follows:
(6)
while P is the average power, U is the effective voltage, and I is the average current. Since this simulation uses a sine wave with a peak value of 1.2 V as the input signal, the value of the effective voltage U is the carrier peak value of V. The transient current waveform at the input of the charge pump is then measured, and then the rms calculation is performed by the internal calculator of cadence to obtain the average current.”
Point 30: Line 230: “average power consumption level of this demodulator is much lower than that of the references”, suggestion: “average power consumption level of this demodulator is much lower than ones reported in the references”
Response 30: I have reworked the statement to make it readable.
Line 279. “is much lower than ones reported in the references.”
Point 31: Line 236: “The average power consumption is 133.9 nA and 113.6 nW”, do you mean 133.9 nW?
Response 31: I apologize for the confusion caused by my poor wording, and for this reason I have removed the value of average current from the article there and only kept the value of average power consumption.
Line 285. “The average power consumption is 113.6 nW”
Point 32: Line 237: “References” not “reference”, line 243: “Table 1 informatively lists the data comparison between this design and the references.” Suggestion: “Table 1 informatively lists the data comparison between this design and the ones of references.”
Response 32: Thank you for the correction, I have fixed the error in the article there.
Line 285. “references”
Line 292. “Table 1 informatively lists the data comparison between this design and the ones reported in references.”
Point 33: Line 238: “the 300 nA power consumption of the demodulator in [7].” 300nW??
Response 33: I apologize that I confused average current with power consumption. I have made a correction in the corresponding place in the article.
Line 286. “The average operating current of this design is 133.9 nA, which is lower than the 300 nA claimed by reference [7].”
Point 34: Line 239: “From the above table, it can be seen…”, which table? There is no table before this line!!
Response 34: I am very sorry that I moved the table and forgot to change the corresponding statement in the article. Thank you very much for your correction and I have corrected the article in that place. I will be more careful when writing future articles.
Line 288. “As can be seen from Table 1,.”
Point 35: Lines 239-241: please insert table confirming the given information: “it can be seen that the present demodulator has a higher and wider transmission bandwidth with a carrier frequency up to 60 times higher than the low carrier frequency used in [1][2][3].”
Response 35: I am very sorry that I moved the location of references [4][5][6] and forgot to change the information cited there. This means that the information is already given in Table 1.
Lines 288 to 290. “As can be seen from Table 1, the demodulator proposed in this paper uses a higher and wider transmission frequency with a carrier frequency approximately 60 times higher than the one claimed by references [4][5][6].”
Point 36: Line 241: could author explain this idea: “The demodulator enables information transmission over long distances with lower power consumption than low frequency carrier designs” because from formula (1) the read range distance increases when wave length increases, this occurs when frequency decreases since . Does author mean a low power consumption design allows a longer read distance than a low frequency design?
Response 36: I am very sorry, my knowledge of the antenna is limited, I dare not jump to conclusions.
Point 37: Line 243: “Table 1 informatively lists the data comparison between this design and the references” suggestion: “Table 1 informatively lists the data comparison between this design and the ones reported in references”
Response 37: I have reworked the statement to make it readable.
Line 293. “Table 1 informatively lists the data comparison between this design and the ones reported in references.”
Point 38: Table 1: 0.3 nA or nW??
Response 38: In Table 1 it is 0.3nA. I have added instructions in the article.
Lines 299 to 301. “It is worth mentioning that in Table 1, the average current is used to characterize the power consumption since the specific voltage values and the specific power consumption values of the demodulator are not given in reference [7].”
Point 39: Line 249, Table1: replace “times” by “years” or “date”
Response 39: I have made the changes in the corresponding places in the table. Thanks again for your guidance on the suggested changes.
Line 298. “Year”
Reviewer 4 Report
This paper proposes a novel architecture for UHF ASK demodulators for passive RFID tags. It achieves its goals and the concept is very appealing. It I well documented and with a great introduction of the concepts involved. Below are some specifics comments:
- The abstract is clear and precise
- The working principle and equations are very well explained.
- (Page 4, line 112-114) Not clear sentence about the signal in the charge pump. I think some rewriting is needed.
- Some graphs for the charge pump operations are required for the sake of providing a clearer picture. Something like what is done with the envelope detector for example. (Page 5)
- The “double” envelope detector for the signal seems an interesting approach.
- (In page 7 from lines 172-184) We can find a nice description of the comparator’s operation. Could you please confirm if the names of the transistors in the text exactly match with the ones show in the picture? Now it can be followed but using same names it can be easier.
- In page 10 lines 235-238, the consumption is shown. It seems to be extremely low which is one of the highlights of the design. In the graph, the periods without signal show a 0nA consumption. There’s no Iq at any branch so far? Seems a bit odd.
- Table 1. Page 10. Please use 'year' instead of 'times'
- Page 11.The figure has no caption. Also, this figure can have maybe other scale for the consumption, as it is too small in comparison with the other showed parameters.
Author Response
Response to Reviewer 4 Comments
Point 1: (Page 4, line 112-114) Not clear sentence about the signal in the charge pump. I think some rewriting is needed.
Response 1: Thank you very much for your recognition of this article and for your valuable comments on it. Also, I apologize for not responding in a timely manner. The following is my revision. It is worth noting that the number of lines I marked may be different from the number of lines in your suggestion because I changed a lot of text in the article.
Lines 127 to 128. “The tuned signal is initially demodulated by a charge pump in order to demodulate the baseband signal from the 900MHz UHF carrier signal.”
Point 2: Some graphs for the charge pump operations are required for the sake of providing a clearer picture. Something like what is done with the envelope detector for example. (Page 5)
Response 2: I measured the voltage changes at the input and output of the charge pump separately and show them in Figure 6. And added some text to explain the picture.
Lines 147 to 149. “Figure 6 shows the results of the charge pump simulation. The pink waveform above is the input signal at 900 MHz, and the blue waveform below is the signal measured at the output of the charge pump.”
Line 152. “
Figure 6. Charge pump simulation results.”
Point 3: (In page 7 from lines 172-184) We can find a nice description of the comparator’s operation. Could you please confirm if the names of the transistors in the text exactly match with the ones show in the picture? Now it can be followed but using same names it can be easier.
Response 3: I have corrected it in the corresponding place in the article.
Line 143. “composed of PM0 and capacitors C0”
Line 146. “composed of PM1 and capacitors C1”
Lines 198 to 207. “the NM1 drain voltage directly reaches the maximum voltage of 1.2 V due to the amplification of about a thousand times, and the PM5 gate voltage is the NM1 drain voltage, and PM5 does not conduct. Since the PM12 drain voltage is about 0.3 V, the NM11 gate is high potential |VGS|>|Vth|, NM11 conducts and the output at the left-OUT terminal is 0 V. For the output of INM influence, NM0 drain voltage is 0.85 V, PM0 gate potential is lower than the source potential, |VGS|>|Vth|, PM0 conducts, thus pulling up NM2, NM5 gate voltage, thus making the two NMOS conductive. Since PM11 drain voltage is 0 V and NM6 gate voltage is 0 V, NM6 does not conduct, making PM2, PM7 gate at high potential, thus PM2, PM7 does not conduct and the right OUT terminal output is 0 V.”
Point 4: In page 10 lines 235-238, the consumption is shown. It seems to be extremely low which is one of the highlights of the design. In the graph, the periods without signal show a 0nA consumption. There’s no Iq at any branch so far? Seems a bit odd.
Response 4: Since the charge pump demodulation gets a part of the voltage waveform at 0V, it leads to no current for part of the time. And exactly what causes this phenomenon, I think this might be a point where I can dig deeper.
Figure 6 may better illustrate the cause of this problem.
Line 152.
Point 5: Table 1. Page 10. Please use 'year' instead of 'times'
Response 5: I have revised the article in the corresponding place.
Line 298. “Year”
Point 6: Page 11.The figure has no caption. Also, this figure can have maybe other scale for the consumption, as it is too small in comparison with the other showed parameters.
Response 6: Another reviewer suggested that I delete the form because it brought no new news. And since the information level gap in Table 2 is too large, I did not find a good solution to make the table look beautiful. On balance, I chose to delete Table 2. and kept only the original Table 1 and Table 3.
Line 299.
Round 2
Reviewer 2 Report
The authors have addressed all my concerns and I feel that the paper has improved significantly.
Author Response
Response to Reviewer 2 Comments
Thank you very much for your approval of this paper. I also appreciate your patience and the patience of the other three reviewers in going through my article and pointing out the shortcomings. During the revision process, I learned how to write the abstract and conclusion better, and also realized that every abbreviation should be used in full the first time it is used. Also, I learned how to describe formulas correctly and use connectives appropriately. I have also added some simulation figures to the text so as to prove my conclusions from different perspectives. Your revision suggestions helped me to understand how to write a good rigorous scientific paper, and I think this is the biggest gain for me. Finally, thank you again for the advice you gave.
